# CRISPR artificial splicing factors

Menghan Du[1,2,6], Nathaniel Jillette[1,6], Jacqueline Jufen Zhu[1], Sheng Li [1,2,3,4] & Albert Wu Cheng [1,2,3,5✉]

Alternative splicing allows expression of mRNA isoforms from a single gene, expanding the diversity of the proteome. Its prevalence in normal biological and disease processes warrant precise tools for modulation. Here we report the engineering of CRISPR Artificial Splicing Factors (CASFx) based on RNA-targeting CRISPR-Cas systems. We show that simultaneous exon inclusion and exclusion can be induced at distinct targets by differential positioning of CASFx. We also create inducible CASFx (iCASFx) using the FKBP-FRB chemical-inducible dimerization domain, allowing small molecule control of alternative splicing. Finally, we demonstrate the activation of *SMN2* exon 7 splicing in spinal muscular atrophy (SMA) patient fibroblasts, suggesting a potential application of the CASFx system.

[1] The Jackson Laboratory for Genomic Medicine, Farmington, CT 06032, USA. [2] Department of Genetics and Genome Sciences, University of Connecticut Health Center, Farmington, CT 06030, USA. [3] The Jackson Laboratory Cancer Center, Bar Harbor, ME 04609, USA. [4] Department of Computer Science and Engineering, University of Connecticut, Storrs, CT 06269, USA. [5] Institute for Systems Genomics, University of Connecticut Health Center, Farmington, CT 06030, USA. [6] These authors contributed equally: Menghan Du, Nathaniel Jillette. ✉email: albert.cheng@jax.org

Splicing is a process in which segments of a pre-mRNA called introns are removed while segments called exons are joined together to form mature mRNA[1]. Alternative splicing is a phenomenon in which different exon segments of a gene are spliced together to form mature mRNA with varying sequences, greatly expanding the protein repertoire coded by a single gene. The process of alternative splicing is deeply embedded in gene regulatory networks and serves to control gene isoform expression of >90% of human genes[2]. Given its prevalence, it is not surprising that dysregulation of RNA splicing has been implicated in many diseases[3–5]. RNA-seq is a powerful tool that can be used to "read" transcriptomes and identify changes in alternative splicing within different cell types, conditions and diseases[2,5,6]. However, a scalable tool for precisely and reversibly "writing" alternative splicing is lacking.

Although isoform-specific RNAi targeting a specific gene isoform for degradation or isoform-specific cDNA over-expression can be used to perturb isoform levels[7,8], the overall expression level of the target gene might not be preserved. While splice-switching antisense oligonucleotides (ASOs) are efficient in perturbing splicing and have even advanced into clinical trials[9], their cost is prohibitive for large scale studies and many designs need to be screened to identify effective target sequences. Also, because ASOs are transient in nature they are not suitable for use cases that require either stable or inducible expression.

Fusion of RNA regulatory proteins to heterologous RNA binding domains, such as Pumilio/PUF, MS2 coat protein (MCP), PP7 coat protein (PCP), and λN, have allowed artificial modulation of RNA processes[10–15]. For example, tethering of serine-rich or glycine-rich domains by engineered PUF domains to exons induce their inclusion or exclusion, respectively[12]. However, these artificial RNA effectors require either protein engineering or insertion of artificial tags to target RNA and depend on short recognition sequences which limits targeting flexibility and specificity.

The fields of genetics and epigenetics have greatly benefited from an explosion of technologies based on RNA-guided DNA-targeting CRISPR-Cas systems[16]. We, among others, have successfully implemented molecular tools for modifying genetic sequences or epigenetic states of target DNA loci[17–25]. CRISPR-mediated DNA-level genetic editing approaches have been used to perturb splicing (base editing/indel at splice sites or cutting out whole exon)[19–21]. However, these may have confounding effects due to potential perturbation of DNA cis-regulatory elements (e.g., transcription factor binding sites) sharing the same piece of DNA. In addition, it is difficult to promote exon inclusion with CRISPR-mediated DNA deletion or mutation methods.

The exciting prospect of using CRISPR to target RNA was first demonstrated by conversion of the most frequently used DNA-targeting SpCas9 to an RNA nuclease "RCas9" with the addition of a PAMmer—an oligo that when bound to target RNA mimics the Protospacer Adjacent Motif (PAM) required for SpCas9 binding[19]. Although targeting of RCas9 to repetitive sequences does not require PAMmer[26], repeat sequences constitute only a small proportion of all RNA cis-regulatory elements. Following the initial report of RCas9, other CRISPR/Cas9 systems were also found to bind to single-stranded RNA in vitro[27,28], but evidence for their in vivo RNA binding in mammalian cells is lacking. RNA-guided RNA nucleases from bacterial CRISPR systems have recently been discovered[29–31]. Their adaptation to mammalian cells has not only allowed programmable RNA degradation[29,31,32] but has also been amenable for engineering new functions, e.g., RNA sequence editing[30], live-cell RNA imaging[32], and diagnostics[33].

In particular, CasRx is the most recently identified type IV-D CRISPR-Cas ribonuclease isolated from *Ruminococcus flavefaciens*

XPD3002 with robust activity in degrading target RNAs matching designed guide RNA (gRNA) sequences[31]. Furthermore, dCasRx with mutated nuclease domains (R239A/H244A/R858A/H863A) can be programmed to bind splicing elements to inhibit exon splicing, potentially by blocking access of splicing machinery. Induction of exon inclusion, however, has yet to be demonstrated. In contrast to exon exclusion that can be sufficiently induced by binding of dCasRx alone[31], induction of exon inclusion might require additional splicing factor activity.

In the present work, we create CRISPR Artificial Splicing Factors (CASFx) by fusing RNA-target Cas proteins with splicing regulatory domains. We show that exon inclusion and exclusion can be induced on two targets simultaneously by positioning CASFx gRNAs on different sequence elements. We further implement a rapamycin-inducible CASFx using the FKBP-FRB dimerization domain to allow small-molecule control of splicing. Finally, we demonstrate activation of *SMN2*-E7 by CASFx in spinal muscular atrophy (SMA) fibroblast cells.

## Results

**CASFx modulates alternative splicing**. We fused dCasRx with splicing factors successfully applied in aptamer tethering assays to activate exon inclusion when bound downstream of the target exon in splicing minigenes[10,14]. Exon 7 of survival motor neuron 2 gene (*SMN2*-E7) was chosen as our test exon as it has implications in SMA treatment, and its regulation is well-characterized[34]. We first constructed CASFx-1 (RBFOX1N-dCasRx-C) by replacing the entire RNA recognition motif (RRM) of splicing factor RBFOX1 (residues 118-189) with dCasRx and tested its ability to induce inclusion of *SMN2*-E7 in an *SMN2* splicing minigene (Fig. 1a). When HEK293T cells were transfected with pCI-SMN2 (containing the splicing minigene) and control GFP plasmid (pmaxGFP), the *SMN2* minigene expressed predominantly exclusion isoform (Fig. 1b, lane 1). To activate *SMN2*-E7 inclusion, four guide RNAs (gRNAs gSMN2-1 through 4) were designed in the intron downstream of *SMN2*-E7 and transfected individually with CASFx-1 and pCI-SMN2 into HEK293T. These resulted in increased *SMN2*-E7 inclusion (Fig. 1b, lanes 11~14, see upper bands). Simultaneous introduction of two, three, or four intronic gRNAs further increased levels of E7-included transcripts and decreased the levels of E7-excluded transcripts, switching the splicing pattern to predominantly inclusion (Fig. 1b, lanes 15~17). *SMN2*-E7 activation is dependent on RBFOX1 effector because dCasRx alone did not result in activation (Fig. 1b, lanes 3~9). Activation is also dependent on the binding of CASFx-1 onto the *SMN2* intron as control gRNAs ("C") did not induce *SMN2*-E7 inclusion (Fig. 1b, lane 10). To test whether dCasRx can tether other splicing factors, we generated two additional CASFx by fusing RBM38 to N- or C-termini of dCasRx, which we called CASFx-2 (RBM38-dCasRx) and CASFx-3 (dCasRx-RBM38), respectively. In addition, since dCasRx binding within exons was shown to induce exon exclusion in a previous study[31], we asked whether CASFx could also be used to induce exon skipping when directed to bind within exons. Therefore, we designed an exonic ("EX") gRNA in the middle of *SMN2*-E7 (Fig. 2a). When co-transfected with one of the CASFx and the pool of intronic downstream SMN2-DN (gSMN2-1,2 and 3), *SMN2* minigene showed a switch to predominantly inclusion isoform (Fig. 2b, lanes 6,9,12). Exon inclusion was dependent on both intronic targeting as well as splicing effector activities, while *SMN2*-E7 exclusion could be induced by exonic targeting of unfused dCasRx or effector-fused CASFx (Fig. 2b, compare lanes 3,5,9,12 and lanes 4,7,10,13), consistent with previous observations[31]. These demonstrate that exon inclusion or exclusion can be induced by the same CASFx by designing gRNA binding to

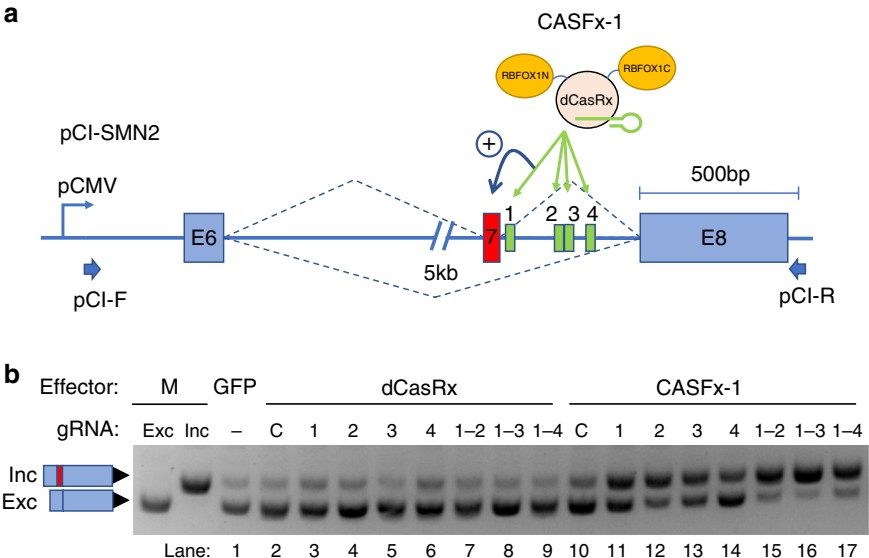

**Fig. 1 Exon inclusion induced by CASFx-1 (RBFOX1N-dCasRx-C). a** Schematic of the CASFx-1 and *SMN2* minigene. The RNA binding domain of RBFOX1 was substituted by dCasRx to create an RNA-guided CRISPR Artificial Splicing Factor 1 (CASFx-1) RBFOX1N-dCasRx-C. The *SMN2* minigene on plasmid pCI-SMN2 contains exons 6 (E6) and 8 (E8) that are constitutively spliced, and exon 7 (E7) that is alternatively spliced, and the intervening introns, driven by the CMV promoter (pCMV). Four designed guide RNA (gRNA) target sites are indicated by numbered boxes 1 through 4 within the intron between E7 and E8. pCI-F and pCI-R indicate primers used for semi-quantitative RT-PCR assays. **b** Gel image of splicing RT-PCR using primers pCI-F and pCI-R on *SMN2* minigene transcripts in cells co-transfected with control GFP plasmid (pmaxGFP), unfused dCasRx, or CASFx-1 (RBFOX1N-dCasRx-C), and the indicated gRNAs, numbers correspond to those in panel a with dash indicating the range of gRNAs used. "C" indicates a control gRNA not matching *SMN2* minigene. Upper and lower bands correspond to the E7-included and -excluded transcripts, respectively. Reference splicing bands derived from E7-excluded sample (pCI-SMN2-MS2) and E7-included sample (pCI-SMN2-MS2 + MCP-RBFOX)[10] serve as molecular weight markers (M) for inclusion (Inc) and exclusion (Exc) events. The image shown is representative of two independent experiments. Uncropped gel images are included in the Source Data file.

different locations on target transcripts. We also quantified splicing changes by quantitative RT-PCR (Fig. 2b, upper column plot). CASFx-1 induced a 21-fold increase while CASFx-2 and CASFx-3 induced ~6-fold increases in inclusion/exclusion relative ratios.

**Examining functional splicing activation domains of CASFx-1.** We next asked which part of RBFOX1 N- and C-terminal domains are required for exon activation by CASFx-1 (RBFOX1N-dCasRx-C). A previous study demonstrated that RBFOX1 truncation mutants lacking the entire RRM domain are functional in inducing cassette exon inclusion[10]. Further truncation experiments demonstrated that the C-terminal domain of RBFOX1 is sufficient for cassette exon activation when targeted to the downstream intron[10]. To identify RBFOX1 domains required in the context of CASFx-1-mediated exon activation, we generated a series of RBFOX1 domain truncation mutants (Supplementary Fig. 1a). While full-length CASFx-1 (RBFOX1N-dCasRx-C) was the most active in promoting exon 7 inclusion (Supplementary Fig. 1b, lane 2), dCasRx-RBFOX1C fusion (C) was also able to induce exon 7 inclusion, albeit to a lesser extent (Supplementary Fig. 1b, lane 4). RBFOX1N-dCasRx fusion (N) was unable to induce splicing changes (Supplementary Fig. 1b, lane 3), consistent with a previous report[10]. Further truncations of the RBFOX1C domain greatly diminished or abolished splicing activity (Supplementary Fig. 1b, lane 5–8), suggesting that the full C-terminal domain is required for CASFx-1-mediated exon activation, and that the inclusion of N-terminal domain enhances splicing activation. It is worth noting that the domain requirement presented here for CASFx-1-mediated splicing activation may be different from that for the endogenous RBFOX1 and should thus be interpreted only in the context of the particular fusion construct.

**Multiplexed splicing modulation by CASFx.** Next, we tested whether more than one splicing event can be modulated simultaneously and differentially with CASFx (Fig. 3a). We targeted CASFx-1 to the splice acceptor site of an RG6 minigene (RG6-SA) and the intron downstream of *SMN2*-E7 minigene and observed simultaneous repression of the RG6 cassette exon (RG6-CX) and activation of *SMN2*-E7 (Fig. 3b, lane 4). Since CasRx is capable of processing gRNAs encoded in tandem in a polycistronic pre-gRNA array by cleaving 5′ of each direct repeat (DR)[31], we asked whether the three SMN2-DN spacers and the RG6-SA spacer could be transcribed as one polycistronic pre-gRNA to achieve simultaneous modulation of the two splicing events. First, we tested if the addition of a preprocessed DR to the 3′ end of gRNA is tolerated by CASFx (Fig. 3a, DR-SMN2-2-DR and DR-RG6-SA-DR). As predicted, these gRNAs remained active in inducing *SMN2*-E7 inclusion and RG6-CX exclusion (Fig. 3b, lanes 5,6). More importantly, a polycistronic pre-gRNA (SMN2-DN-RG6-SA) harboring the three SMN2-DN spacers and the RG6-SA spacer induced simultaneous *SMN2*-E7 inclusion and RG6-CX exclusion when transfected with CASFx (Fig. 3b, lane 7), confirming the functionality of the polycistronic pre-gRNA architecture in inducing simultaneous and bidirectional splicing modulation of two different targets.

**Comparison of CASFx with PUF-based engineered splicing factor.** Previously, engineered splicing factors (ESFs), based on the fusion of splicing factors with programmable RNA-binding PUF domains (PUF-ESFs), have been reported to modulate alternative splicing[12]. Here we compared the efficiency and specificity of CASFx with PUF-based ESF (PUF-ESF). To do this we assembled three PUF-ESF modules designed to bind to the same locations as the three CASFx-SMN2-gRNA target sequences (Fig. 4a). Transfection with one of the three PUF-ESFs induced

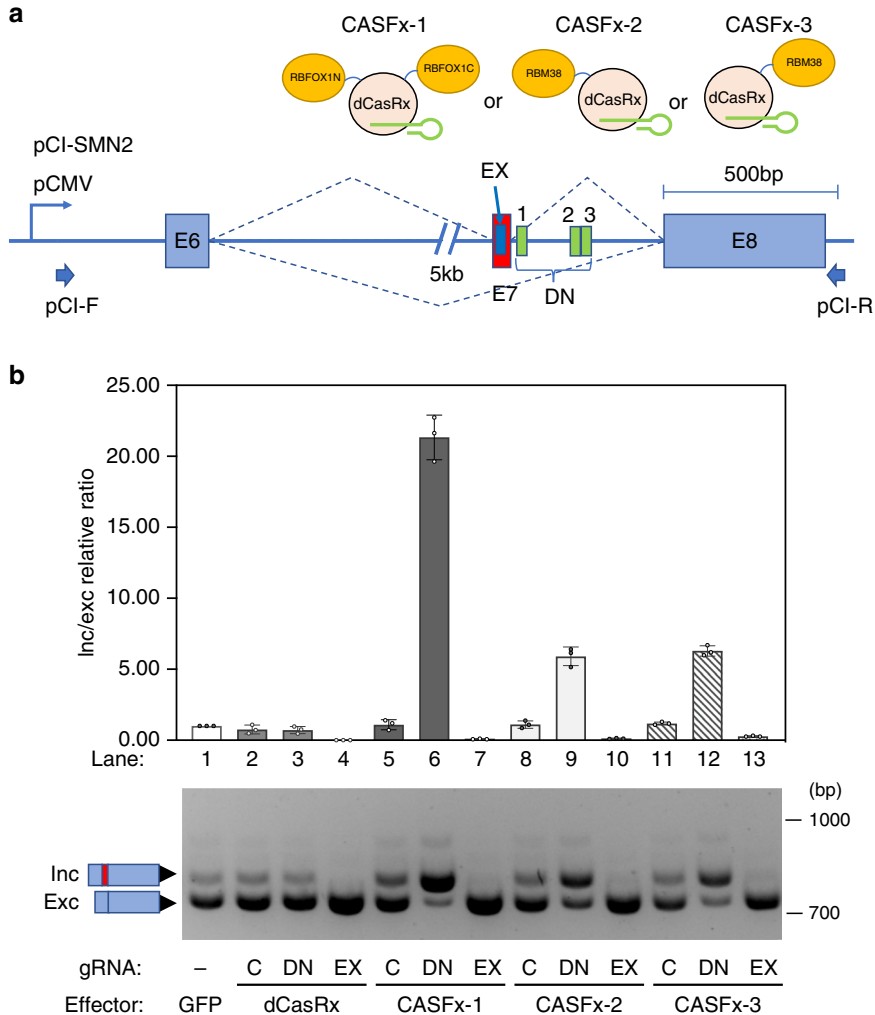

**Fig. 2 Activation and repression of exon by differential positioning of CASFx. a** Schematic of the CRISPR artificial splicing factors, CASFx-1 (RBFOX1N-dCasRx-C), CASFx-2 (RBM38-dCasRx), CASFx-3 (dCasRx-RBM38) and *SMN2* minigene, as well as a set of three target sites downstream of E7 (DN: gRNA-1 through 3) and one target site target within E7 (EX). **b** Upper panel shows the inclusion/exclusion (inc/exc) ratio fold-change assayed by qRT-PCR on *SMN2* minigene transcripts in cells co-transfected with dCasRx, CASFx, and the indicated gRNAs, with respect to the GFP control (set to 1). Data are represented as mean ± SD (*n* = 3). Lower panel shows a gel image of semi-quantitative splicing RT-PCR of the corresponding samples. "C" indicates a control gRNA without matching *SMN2* minigene sequence; "DN" indicates a pool of three gRNAs (SMN2-gRNA-1 through 3) targeting downstream of E7; "EX" indicates a gRNA targeting within E7. Uncropped gel images and qRT-PCR values are included in the Source Data file.

*SMN2*-E7 inclusion (Fig. 4b, lanes 2~4). The co-transfection of the three PUF-ESFs did not provide an additive effect on exon 7 inclusion compared with PUF-ESF3-only transfection (Fig. 4b, lane 5). As we show above, cells transfected with separate CASFx effectors and gRNA plasmids also showed an increase of the inclusion isoform and a decrease of the exclusion isoform (Fig. 4b, lanes 14, 17 and 20). We also constructed two all-in-one CASFx constructs expressing both CASFx effectors and three gRNAs (Fig. 4a). CASFx all-in-one plasmids provided the most efficient splicing induction (Fig. 4b, lane 7 and 9). Quantification by RT-PCR revealed a 14-fold increase in the inc/exc relative ratio with the CASFx-1 all-in-one sample and more than a 7-fold increase with the CASFx-3 all-in-one sample. Of the three PUF-ESF tested, PUF-ESF3 had the strongest induction with an approximately 10-fold increase (Fig. 4b lane 2~4). These data demonstrate that splicing activation offered by CASFx is on-par with PUF-ESF.

Off-target effects are a major concern for any sequence-based gene editing methods. We therefore investigated the genome-wide specificity of CASFx and PUF-ESF using RNA-seq. Four

off-target splicing changes were induced by CASFx-1 and two were induced by CASFx-3 (Fig. 4c i and ii). For cells transfected with PUF-ESFs, analysis of alternative splicing changes revealed 59 off-target splicing changes (Fig. 4c iii). These results demonstrate that CASFx-1 and CASFx-3 offer higher specificity compared with PUF-ESF. One caveat for such interpretation, however, is that the higher off-target effects of PUF-ESF could also result from the relatively higher molar amount of the three PUF-ESFs transfected as separate plasmids. No significant changes in endogenous *SMN2* splicing were detected by RNA-seq (Fig. 4c). It is because the high sequence homology between *SMN1* and *SMN2* and the constitutive inclusion of *SMN1* exon 7 in HEK293T preclude accurate quantification of endogenous *SMN2*-E7 splicing level.

For CASFx, off-target effects may result from mismatch tolerance between the gRNA and the target RNA. To investigate the contribution of mismatch tolerance to the observed CASFx off-targets, we generated sequence match profiles of the three gRNA spacers against sequences encompassing the upstream intron, cassette exon, and downstream intron of the off-target

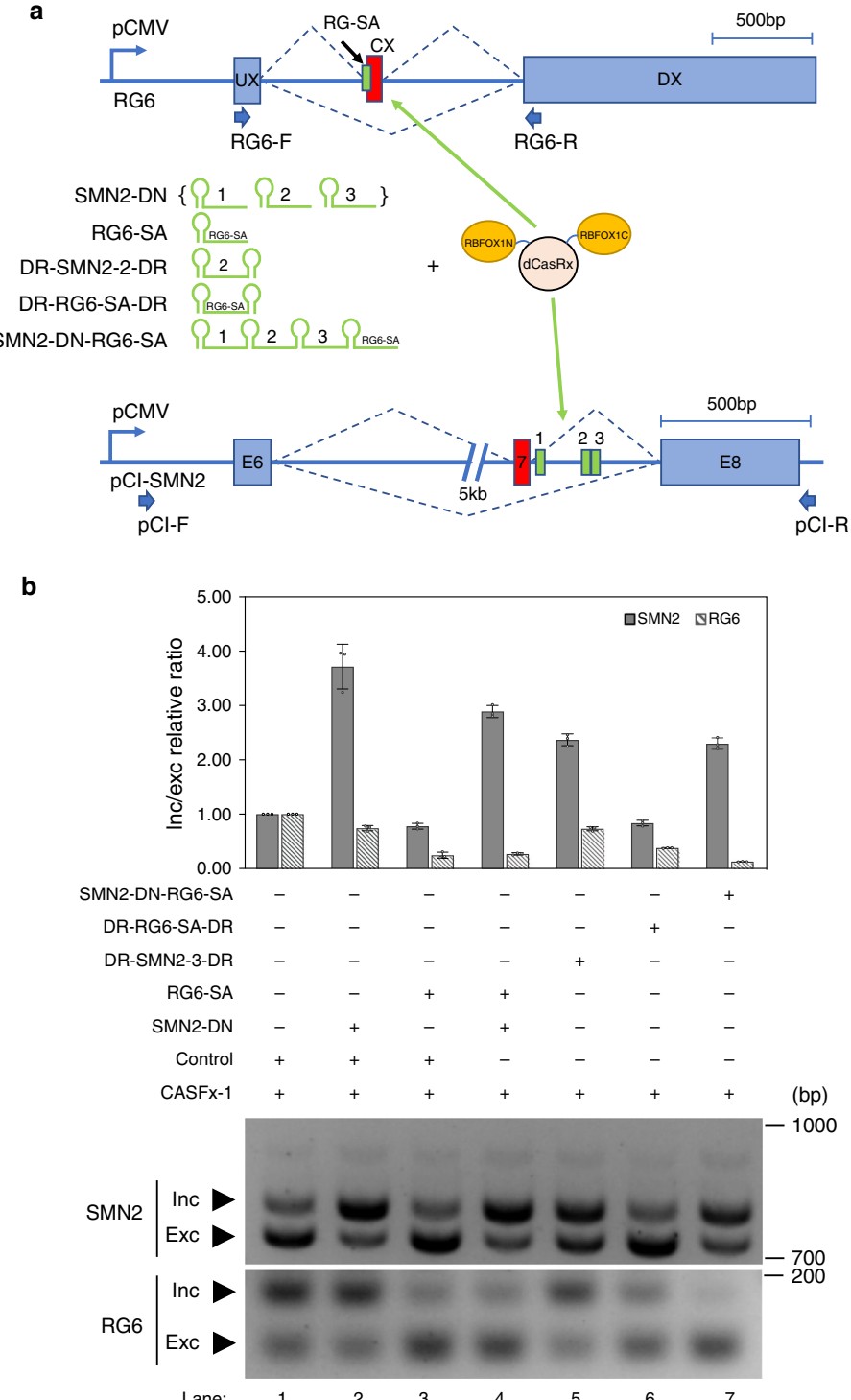

**Fig. 3 Simultaneous activation and repression of two independent exons by RBFOX1N-dCasRx-C. a** Schematic of the CASFx-1, various gRNA architectures, as well as the RG6 and *SMN2* minigenes. SMN2-DN gRNAs is a pool of three gRNAs (SMN2-gRNA-1 through 3), each expressed by a separate plasmid, targeting the corresponding numbered locations on the *SMN2* minigene. RG6-SA targets splice acceptor of RG6 cassette exon (CX). DR-SMN2-2-DR is *SMN2* target gRNA 2 flanked by two direct repeats (DR). DR-RG6-SA-DR contains spacer against RG6-CX splice acceptor flanked by two DRs. SMN2-DN-RG6-SA is a polycistronic pre-gRNA with spacers targeting three DN sites on *SMN2* downstream intron and RG6-CX splice acceptors intervened by DRs. **b** Upper panel shows inclusion/exclusion (inc/exc) ratio fold-changes assayed by qRT-PCR on *SMN2* minigene transcripts in cells co-transfected with the two minigene plasmids, CASFx-1 (RBFOX1N-dCasRx-C) and the indicated gRNAs. *SMN2* and RG6 splicing changes are represented by gray and slash pattern filled bars, respectively. Fold-changes are relative to cells transfected with GFP control (set to 1). Data are represented as mean ± SD (*n* = 3). Lower panel shows a gel image of semi-quantitative splicing RT-PCR of RG6 and *SMN2* minigene transcripts in cells co-transfected with the two minigene plasmids, CASFx-1 and the indicated gRNAs. Uncropped gel images and qRT-PCR values are included in the Source Data file.

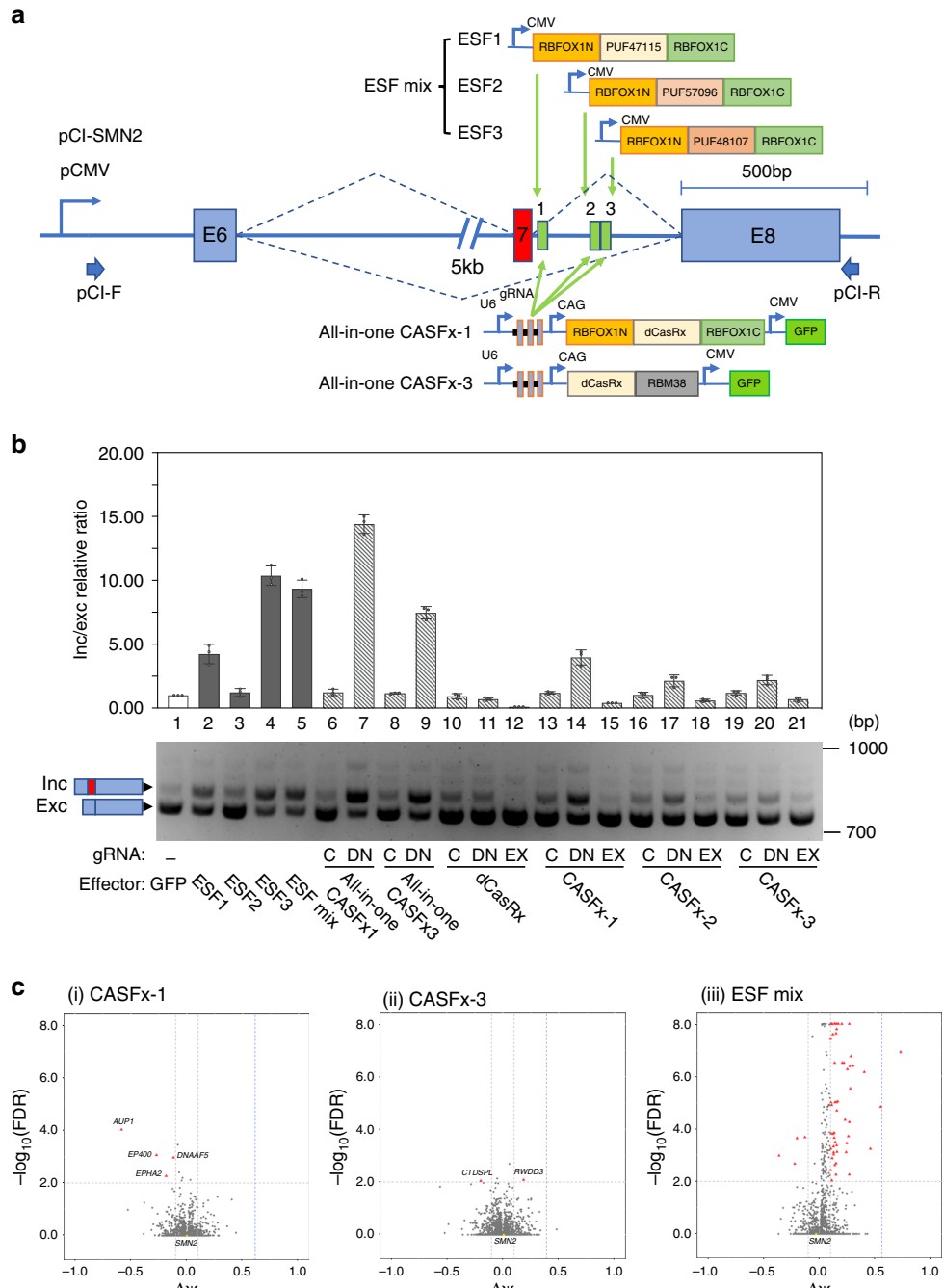

**Fig. 4 Efficiency and specificity comparison of exon activation induced by CASFx and PUF-ESF. a** Schematic of the three PUF-based engineered splicing factors (PUF-ESFs), two all-in-one CASFx-1 plasmids and the target sites of gRNA 1-3 on the *SMN2* minigene. ESF mix represents a pool of three PUF-ESF constructs. **b** Upper panel shows inclusion/exclusion (inc/exc) ratio fold-changes assayed by qRT-PCR on *SMN2* minigene transcripts in transfected cells with respect to GFP control (set to 1). Data are represented as mean ± SD ($n = 3$). Lower panel shows a gel image of semi-quantitative splicing RT-PCR of *SMN2* minigene transcripts in cells transfected with indicated constructs. **c** Plots showing changes in percentage spliced-in ($\Delta\Psi$) and rMATS-computed Benjamini-Hochberg False Discovery Rate (FDR) of cassette exons in cells transfected with indicated constructs compared with GFP control, as determined by RNA-seq ($n = 2$ biological replicates per sample). Exons with significant changes ($|\Delta\Psi| \geq 0.1$ and FDR $\leq 0.01$) are colored in red. $\Delta\Psi$ of *SMN2*-minigene was calculated based on the RNA-seq reads matching with minigene-specific sequences and is indicated as blue vertical dotted line. Uncropped gel images and qRT-PCR values are included in the Source Data file.

events (Supplementary Figs. 2 and 3, Supplementary Table 3). Of the observed off-targets, the closest matches (16 nt) between gRNA and off-target RNA are located on the intronic regions of *DNAAF5* and *CTDSPL* (Supplementary Figs. 2 and 3). To avoid the potential off-target effects in future studies, gRNA designs should consider imperfect transcriptomic matches.

Since RBFOX1 and RBM38 are tissue-specific splicing factors, we asked whether their splicing effector activity could be extended to other cell types when tethered by dCasRx. We tested CASFx-1 and CASFx-3 for activating *SMN2*-E7 in HeLa and U2OS cell lines. As expected, targeting of these CASFx at intronic or exonic regions resulted in the inclusion and exclusion of

*SMN2*-E7 on *SMN2* minigene respectively (Supplementary Fig. 4). Utilizing similar design principles, we used CASFx-1 and CASFx-3 to rescue a minigene cassette exon splicing of a bona fide RBFOX target, *NUMA1*, in an RBFOX2-knockdown HeLa and U2OS cells (Supplementary Fig. 5). These results show that CASFx composed of tissue-specific splicing factors can be applied in different cell types, and can be used for studying the functions of splicing factors and individual target exons.

**Inducible CASFx platform.** To enable tunable control for CASFx, we created two-peptide inducible CRISPR Artificial Splicing Factors (iCASFx) using rapamycin-inducible FKBP-FRB chemically inducible domains (CID)[35]. We separated the RNA binding module (FKBP-dCasRx, or dCasRx-FKBP) and exon splicing effector module (RBFOX1N-FRB-C), into two peptides that can be induced to interact via the FKBP-FRB domains in the presence of rapamycin (Fig. 5a). Induction of *SMN2*-E7 inclusion was observed in cells cultured with rapamycin and transfected with iCASFx plus SMN2-DN-gRNA plasmids (Fig. 5b, lanes 3 and 8), inducing more than a 4-fold increase in inc/exc relative ratio by FKBP-dCasRx and 3.8-fold increase by dCasRx-FKBP. *SMN2*-E7 was not induced when RBFOX1N-FRB-C module was replaced by FRB-Clover (Fig. 5b, lane 6 and 11), demonstrating that rapamycin, dCasRx target binding, and RFBOX1/RBM38 effector domains are all required for the inducible splicing modulation. When cells were co-transfected with iCASFx and control gRNA or RG6-SA, a slight increase of the inclusion isoform was observed (Fig. 5b, lanes 4, 5, 9 and 10). This non-specific increase might have resulted from the non-specific binding of the iCASFx that was not observed in the constitutive CASFx constructs, pointing to the need for additional optimization of the inducible system.

**CASFx-mediated modulation of endogenous *SMN2* splicing.** Having demonstrated the success of CASFx in regulating alternative splicing on minigenes, we next investigated whether CASFx could be applied to endogenous transcripts. In addition to dCasRx-based CASFx, we were interested to see if CASFx could also be constructed using catalytically inactive dPspCas13b, derived from a Cas13b effector from *Prevotella sp. P5-125*[30]. To this end, we swapped dPspCas13b into CASFx-1 in place of dCasRx, generating RBFOX1N-dPspCas13b-C (Fig. 6a). We tested the efficiency of both CASFx constructs in activating endogenous *SMN2*-E7 inclusion in GM03813 fibroblast cells derived from a type ll SMA patient. The homozygous deletion of exons 7 and 8 in the *SMN1* gene allows us to quantify the splicing of endogenous *SMN2*-E7 unambiguously. GM03813 cells were nucleofected with an all-in-one plasmid carrying either CASFx-1 or RBFOX1N-dPspCas13b-C, a polycistronic three-gRNA array, and a GFP marker. To isolate cells that have successfully taken up the plasmids, we collected GFP positive cells via fluorescence-activated cell sorting (FACS) 5 days after nucleofection. Subsequently, RT-PCR was applied to examine the extent of exon 7 inclusion of endogenous *SMN2* transcripts. The expression level of the exon7-included *SMN2* isoform was promoted only in cells transfected with both CASFx and SMN2-gRNAs (Fig. 6b, lane 3 and 5) and showed more than a 2-fold increase in inc/exc relative ratio by CASFx-1 and a 2.75-fold increase by RBFOX1N-dPspCas13b-C. These results not only demonstrate the application of CASFx for modulating endogenous RNA splicing in disease-relevant cell models, but also suggest that orthogonal RNA-targeting CRISPR-Cas systems can be used to tether different splicing factors to regulate splicing processes, providing options for engineering expanded libraries of artificial splicing and RNA regulatory effectors.

## Discussion

In this study, we reported the development of CASFx, artificial splicing factors based on RNA-targeting CRISPR-Cas systems[30,31]. CASFx with RBFOX1 or RBM38 fusions can induce exon inclusion when targeted to bind at a downstream intron, and induce exon exclusion when guided to bind within a target exon. We also showed that simultaneous exon inclusion and exclusion can be achieved by a pool of separate gRNAs or a polycistronic pre-gRNA with multiple target-specific spacers. To augment the controllability of the system, we engineered a rapamycin-inducible CASFx (iCASFx), that allows small-molecule control of alternative splicing. To demonstrate the proof-of-principle application for studying alternative splicing in diseases, we used CASFx to modulate clinically relevant *SMN2*-E7 inclusion in SMA patient fibroblasts.

CASFx offers many advantages compared with other strategies for regulating alternative splicing. As we have demonstrated in this study, CASFx provides higher specificity compared with PUF-ESF and targeting can be easily programmed by changing gRNAs without the need to engineer a new protein for each unique target sequence, as is the case with PUF-ESF. Moreover, unlike RNA-targeting with RCas9[26], dCasRx does not require an additional PAMer oligo. In contrast to CRISPR/Cas9-mediated genome editing strategies for perturbing splicing cis-regulatory elements[36–38], CASFx does not introduce permanent changes in the genome and can be applied transiently and presumably reversibly. Another main method to regulate alternative splicing is to use ASO paring with specific regions of the pre-mRNA of interest. Here too, CASFx offers advantages over the short-lived effects of ASOs because CASFx can use various delivery vectors for either transient or stable expression[39]. By incorporating FKBP-FRB rapamycin-inducible domains, inducible CASFx (iCASFx) allows an additional layer of splicing control via small molecules. Taken together our results show that CASFx broadens the CRISPR toolkit for studying and regulating alternative splicing and represents a promising therapeutic solution for RNA mis-splicing diseases.

## Methods

**Cloning.** HEK293T cDNA was used as a source for PCR-amplification of coding sequences of splicing factors or other RNA binding proteins. Alternatively, gBlocks encoding human codon optimized versions of their coding sequences were ordered from IDT to serve as the PCR template. The pXR002: EF1a-dCasRx-2A-EGFP 28 plasmid (Addgene #109050) served as PCR template for dCasRx coding sequence. The pC0039-CMV-dPspCas13b-GS-ADAR2DD(E488Q) plasmid (Addgene #103849) served as the template for dPspCas13b. The coding sequences of the CRISPR Artificial Splicing Factors (CASFx) were then cloned into pmax expression vector (Lonza) by a combination of fusion PCR, restriction-ligation cloning and Sequence- and Ligation-Independent Cloning (SLIC)[40]. gRNA expression cloning plasmids were generated by similar procedures using IDT oligonucleotides encoding CasRx gRNA direct repeat and PCR reaction using a ccdbCam selection cassette (Invitrogen) and a U6-containing plasmid as templates. Two BbsI restriction sites flanking the ccdbCam selection cassette serves as the restriction cloning sites for insertion of target-specific spacers. Target-specific spacer sequences were then cloned into gRNA expression plasmids by annealed oligo-nucleotide ligation. gRNA sequences are listed in supplementary Table 1. Plasmid listing is included in the supplementary information. Plasmids and Genbank files will be available on Addgene. Additional information is available on http://cheng.bio/casfx

**Cell culture and transfection.** HEK293T (ATCC), U2OS (ATCC), HeLa (ATCC) and the SMA patient-derived fibroblast line GM03813 (Coriell Institute) cells were cultivated in Dulbecco's modified Eagle's medium (DMEM) (Sigma) with 10% fetal bovine serum (FBS)(Lonza), 4% Glutamax (Gibco), 1% Sodium Pyruvate (Gibco) and penicillin-streptomycin (Gibco). Incubator conditions were 37 °C and 5% CO2. For activation experiments, cells were seeded into 12-well plates at 100,000 cells per well the day before being transfected with 600 ng (the "quota") of plasmid DNA with 2.25 μL Attractene transfection reagent (Qiagen). Eighteen nanograms of each reporter minigene plasmid was transfected. The remaining quota was then divided equally among the effector and gRNA plasmids. In cases where there were two or more gRNA plasmids, the quota allocated for gRNA plasmids is further

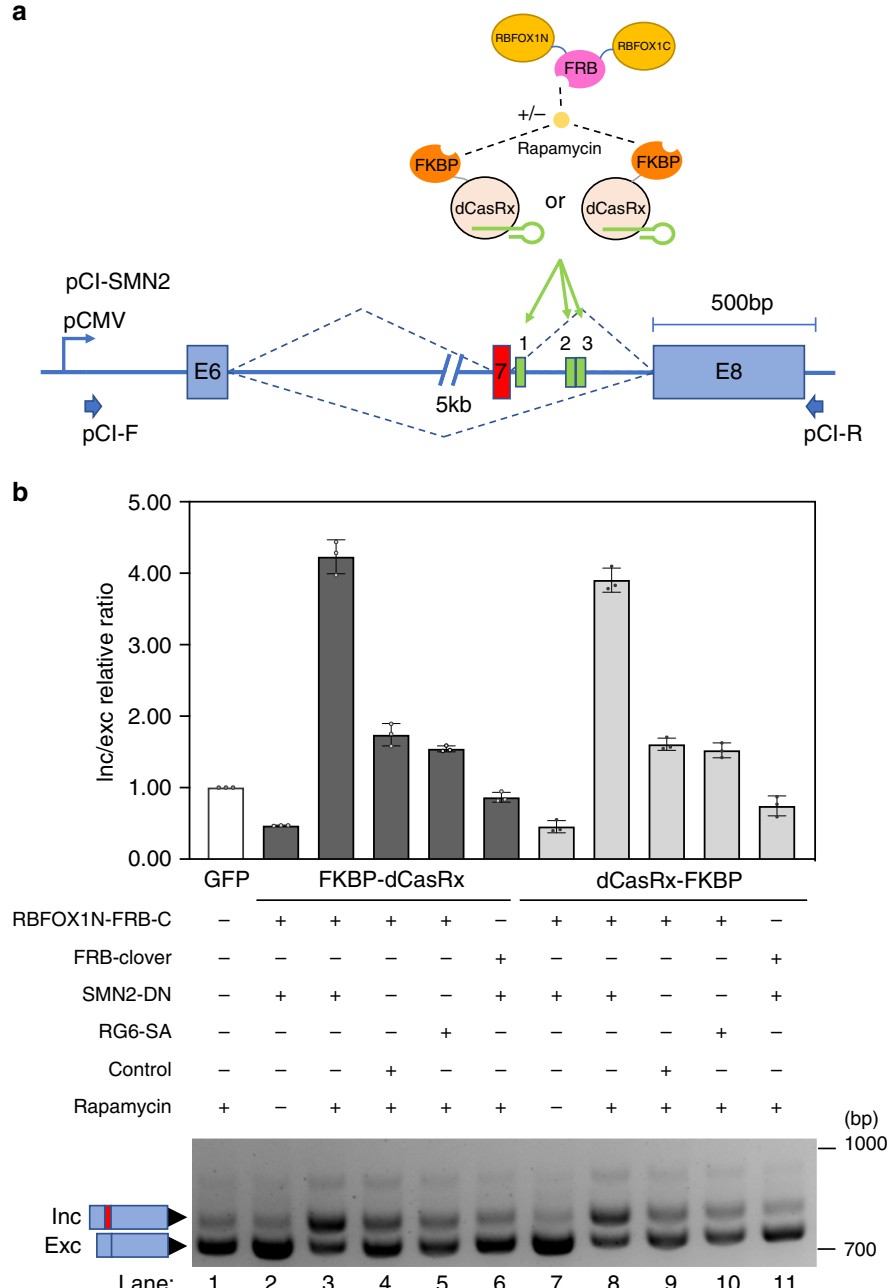

**Fig. 5 Chemically inducible exon activation by iCASFx. a** Schematic of the two-peptide CRISPR artificial splicing factors inducible by rapamycin. The RNA binding module (FKBP- dCasRx or dCasRx-FKBP) and effector module (RBFOX1N-FRB-C) containing the splicing activator domains are expressed separately as two peptides, fused to FKBP or FRB, respectively. FKBP and FRB can be induced to interact by rapamycin, bringing together the RNA binding module and the splicing activator module, and when guided by gRNAs, assemble at the target to activate exon inclusion. **b** Upper panel shows inclusion/exclusion (inc/exc) ratio fold-changes assayed by qRT-PCR on *SMN2* minigene transcripts in cells co-transfected with the indicated constructs, and cultured with ("+") or without ("−") rapamycin. Fold-changes are relative to GFP-transfected sample. Data are represented as mean ± SD (n = 3). Lower panel shows a gel image of semi-quantitative splicing RT-PCR of *SMN2* minigene transcripts in cells co-transfected with the indicated constructs, and cultured with ("+") or without ("−") rapamycin. Uncropped gel images and qRT-PCR values are included in the Source Data file.

subdivided equally. For ESF effectors, an equally split amount (194 ng) of all three ESF plasmids were co-delivered as an ESF mix. For two-peptides effectors (i.e., the FKBP-FRB systems), the effector plasmid quota was divided equally between the plasmids encoding the individual peptides (146 ng). Media was changed 24 h after transfection. Rapamycin (100 nM final concentration) was added during the media change if applicable. Cells were harvested 48 h after transfection for RT-PCR analysis.

**SMA patient cell line nucleofection**. Plasmid delivery by nucleofection was performed using a Nucleofection X unit with the Nucleofection P2 kit (Lonza). For each reaction, 1 × 10⁶ cells were harvested and centrifuged at 100 × *g* for 3 min. The

cell pellet was then re-suspended carefully in P2 Solution and mixed together with 6 μg plasmid. EN-150 program was applied for nucleofection. Five days after nucleofection, cells were trypsinized, suspended in media then sorted on The BD FACSAria™ III sorter (BD Bioscience) to collect GFP positive cells.

**RT-PCR**. Cells were harvested for RNA extraction using RNeasy Plus Mini Kit (Qiagen). Equal amounts of RNAs from one transfection experiment (either 700 or 1000 ng) were reverse-transcribed using a High Capacity RNA-to-cDNA Kit (ThermoFisher). PCR was then performed using 2-μL cDNA using Phusion® High-Fidelity DNA Polymerase (NEB) using minigene plasmid-specific primers for 25

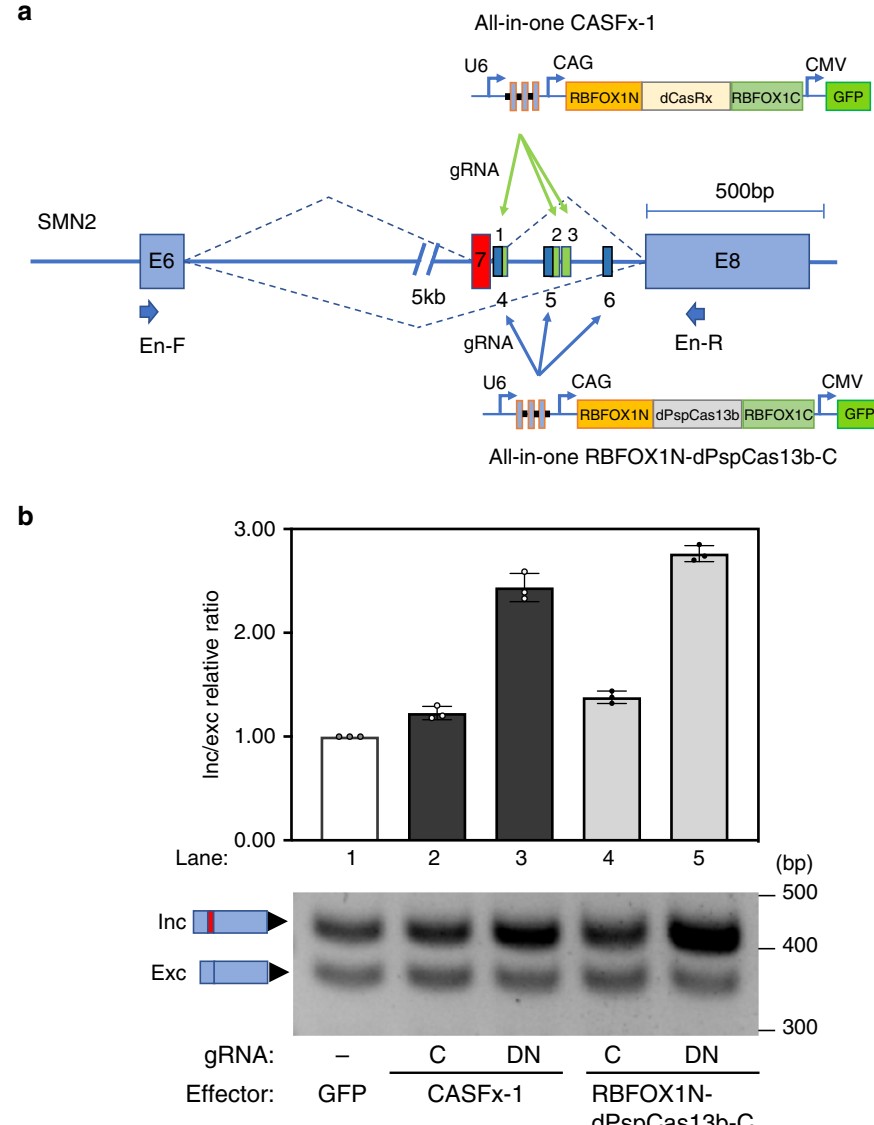

**Fig. 6 Exon activation in SMA patient cells by CASFx. a** Schematic of the two all-in-one CASFx constructs (CASFx-1: RBFOX1N-dCasRx-C and RBFOX1N-dPspCas13b-C) and six gRNAs targeting the corresponding numbered locations on the endogenous intron. **b** Upper panel shows inclusion/exclusion ratio fold-changes of the endogenous *SMN2* gene in patient cells assayed by qRT-PCR 5 days after nucleofection with the indicated CASFx and gRNA relative to GFP control (set to 1). Data are represented as mean ± SD (*n* = 3). Lower panel shows a gel image of semi-quantitative splicing RT-PCR using primers En-F and En-R on endogenous *SMN2* transcripts in the indicated samples. Uncropped gel images and qRT-PCR values are included in the Source Data file.

cycles. The primer sequences are provided in Supplementary Table 2. PCR products were then analyzed on a 3% agarose gel.

**Quantitative RT-PCR**. Quantitative RT-PCR reactions were performed using a LightCycler® 480 instrument (Roche) and the amplifications were done using the SYBR Green PCR Master Mix (Roche). Biological and technical replicate reactions were performed using 3 μL of diluted cDNA in a 10 μL reaction. All primer sequences used are provided in Supplementary Table 2. The relative ratio of inclusion/exclusion isoform was calculated using the ΔΔCt method and presented as a fold change compared with WT cells.

**RNA-seq and data analysis**. Illumina Hiseq 2 × 150 bp sequencing was performed by GENEWIZ. PolyA selection was applied for mRNA species in library preparation. FASTQ raw sequence files were quality checked with FASTQC version 0.11.5[41]. Low quality reads and adapter sequences were trimmed by Trimmomatic version 0.33[42]. Due to the length requirement of the downstream multivariate analysis of transcript splicing (rMATS)[43], reads were processed to match the same length of 135 bp based on the QC report. Shorter reads were discarded and longer reads were trimmed to remove the poor quality 3′ end. Mapping was performed by STAR aligner version 2.5.3 to the human genome (USCS RefSeq hg38 annotation)[44]. We then identify

transcriptome-wide splicing events by rMATS version 3.2.5 using UCSC RefSeq hg38 GTF file as annotation. Each group was compared with the negative control transfected with GFP plasmid alone (*n* = 2). The inclusion level difference (Δψ) of each candidate exon skipping (SE) event was calculated using reads mapping to the body of exons as well as to splice junctions. Considering that low coverage exons and splicing junctions lead to low confidence inclusion levels, we filtered out the cases in which average counts of two replicates for inclusion or skipping were <10. To discover off-target SE event, we set the threshold parameters at |Δψ| ≥ 0.1 and false discovery rate (FDR) ≤ 0.01. For the targeted *SMN2* minigene, inclusion level difference (Δψ) was estimated based on minigene reads extracted from raw FASTQ files containing 8 bp vector-specific sequences. Inclusion or exclusion isoforms of the targeted minigene was then counted using these minigene reads based on their junction site sequences.

**Reporting summary**. Further information on research design is available in the Nature Research Reporting Summary linked to this article.

**Data availability**
The authors declare that the data that support the findings of this study are included in the published article and in the Supplementary Information, and are available from the

corresponding author upon reasonable request. Data containing RNA-seq raw sequencing read files were deposited onto sequencing read archive (SRA) with accession number PRJNA624911 [https://www.ncbi.nlm.nih.gov/bioproject/PRJNA624911/]. Plasmids are available on the Addgene repository [https://www.addgene.org/browse/article/28196786/]. Uncropped gel images presented in all figures, rMAT result tables, qRT-PCR values are provided as a Source Data file. Source data are provided with this paper.

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

## Acknowledgements

This work has been supported by the Jackson Laboratory internal grants (to A.C.), National Human Genome Research Institute 1R01HG009900 (to A.C.) and National Cancer Institute P30CA034196 (to A.C.), Leukemia Research Foundation New Investigator Grant (to S.L.), The Jackson Laboratory Director's Innovation fund 19000-17-31 (to S.L.), The Jackson Laboratory Cancer Center New Investigator Award (to S.L.), and National Cancer Institute of the National Institutes of Health P30CA034196 (to S.L.). We thank the Jackson Laboratory Flow Cytometry for FACS experiments, and Jackson Laboratory Research Program Development for help with manuscript editing.

## Author contributions

M.D., N.J., and A.W.C. conceived and designed the study. M.D., N.J., J.J.Z., and A.W.C. performed the experiments. M.D., N.J., S.L., and A.W.C. analyzed data and wrote the manuscripts.

## Competing interests

A provision patent application (WO2020069331) by N.J. and A.W.C. describing the invention has been filed. All other authors declare no competing interests.
