## [Peer Review File · Nature Communications]

Reviewers' Comments:

Reviewer #1:

Remarks to the Author:

This manuscript from Jillette and Cheng describes the use of dCasRx (a type IV-D CRISPR-Cas ribonuclease with nuclease domains mutated) fused to splicing factor domains to modulate splicing in mammalian cells. This work builds on that using designer PUF proteins, RCas9, and dCasRx to modulate mRNA splicing. Compared to the latter work using dCasRx, this is distinct as it uses RBFOX1 or RBM38 splicing domains instead of hnRNPa1 and also added the ability to chemically control splicing modulation via FKBP-FRB domains. However this is really an incremental advance on the previously described systems. Furthermore, the experimental work is somewhat superficial with non-quantitative RT-PCR as the only assay used to assess the effectiveness of the dCasRx-fusions. In addition, the only mRNA examined was one produced from a transfected, CMV promoter-driven mini-gene.

To be a meaningful addition to the literature the authors should:

1. Provide a side-by-side comparison of their system to the existing benchmark tools for splicing modulation, e.g. to Zefeng Wang's PUF-based engineered splicing factors.
2. Test the effectiveness of their system against mRNAs produced from endogenous genes.
3. Check for transcriptome-wide off targets using isoform-specific analyses of RNA-Seq data using their system compared to negative controls and benchmark splicing modulators.

Without these experiments it isn't clear how well the authors' system actually works, which will impede its uptake by other researchers. Therefore, this study requires significantly more development before publication.

Reviewer #2:

Remarks to the Author:

The authors of this manuscript invented CASFx, artificial splicing factors by composing domains of RESPR and RNA-binding proteins RBFOX1/RBM38. The authors guided the chimera splicing factor to ISSs located in the intron of SMN2 and E7, and examined the function of the artificial splicing factors by RT-PCR. They further developed this to inducible CASFx (iCASFx) which can be induced by Rapamycin. Although their idea seems to be attractive, the data are too primitive to convince me. More careful study would be required before the publication as pointed below.

- 1) In Fig 1, RBFOX1N of the chimera splicing factor may activate E7 recognition just by nonspecifically interfering the binding of hnRNPs to ISSs. The authors should confirm that the RBFOX1N chimera splicing factor activated E7 by binding to the ESS in E7 and that which part of RBFOX1N is required for the activation.
- 2) In Fig2, dCasRx alone promotes E7 skipping suggesting that dCasRx-binding itself may interfere with U1 or U2 binding at either the donor site or acceptor site, because E7 is very short (only 44 nucleotides).
- 3) In Fig3, it is not surprised that binding of dCasRx to the splicing acceptor site suppressed the CX exon inclusion, though the RT-PCR data of RG6 is not so clear in comparison with SMN2. The authors should show the data of dCasRx without RBFOX1N as control. In addition, gRNA of DR-SMN2-3-DR in panel B should be gRNA of DR-SMN2-2-DR.
- 4) Though Rapamycin-dependent splicing regulation is an interesting idea, it is not sure that the activity of RBFOX1/RBM38 is really regulated by Rapamycin. They need to show the data of an FRB-conjugated protein as control.
- 5) The authors should show the sequence data of target of sgRNA in exon 7 and intron 7. I could not find them at the addgene site.
- 6) All the data just come from the transient transfection to HEK293T and RT-PCR. The authors should confirm their data in other cell lines. In addition, they should test whether their chimera splicing factors can regulate splicing of endogenous pre-mRNAs.

7) They should examine the function of their chimera splicing factors targeting splicing of RBFOX- and RBM38-dependent exons. By replacing the cis-element with sgRNA would be contribute to prove the function of their artificial splicing factors.

8) If the authors will show that their chimera splicing factors can regulate any alternative splicing using in vitro splicing system, they would convince us more about their chimera splicing factors

Response to reviewers' comments (NCOMMS-18-36967)

First of all, we appreciate the reviewers' time and feedback. We have added substantial data to support and improve the manuscript according to the valuable suggestions offered by the two reviewers. Here we would like to explain changes to the manuscript point-by-point below:

Reviewer #1 (Remarks to the Author):

This manuscript from Jillette and Cheng describes the use of dCasRx (a type IV-D CRISPR-Cas ribonuclease with nuclease domains mutated) fused to splicing factor domains to modulate splicing in mammalian cells. This work builds on that using designer PUF proteins, RCas9, and dCasRx to modulate mRNA splicing. Compared to the latter work using dCasRx, this is distinct as it uses RBFOX1 or RBM38 splicing domains instead of hnRNPa1 and also added the ability to chemically control splicing modulation via FKBP-FRB domains. However, this is really an incremental advance on the previously described systems. Furthermore, the experimental work is somewhat superficial with non-quantitative RT-PCR as the only assay used to assess the effectiveness of the dCasRx-fusions. In addition, the only mRNA examined was one produced from a transfected, CMV promoter-driven mini-gene. To be a meaningful addition to the literature the authors should:

1. Provide a side-by-side comparison of their system to the existing benchmark tools for splicing modulation, e.g. to Zefeng Wang's PUF-based engineered splicing factors.

Thank you for your suggestion to set up a comparison between CASFx and PUF-based engineered splicing factors (PUF-ESF). We have now added the comparison between PUF-ESF and CASFx, to assess their induction efficiency and specificity. We compared the splicing induction efficiency via semi-quantitative and quantitative RT-PCR. Though efficiency given by each CASFx is slightly different, CASFx-1 (RBFOX1-dCasRx-C) showed the highest fold-change of inclusion/exclusion relative ratio, slightly higher activity than PUF-ESF (Fig 4B). Based on RNA-seq results, CASFx showed higher specificity with far fewer transcriptome-wide off-target events than PUF-ESF (Fig 4C).

2. Test the effectiveness of their system against mRNAs produced from endogenous genes.

We have tested CASFx in spinal muscular atrophy patient cells (GM03813) against pre-mRNA of endogenous SMN2 gene. The results showed that both dCasRx- and dPspCas13b-based CASFx were able to induce the increasing of SMN2-FL, the exon 7 included isoform (Fig 6).

3. Check for transcriptome-wide off targets using isoform-specific analyses of RNA-Seq data using their system compared to negative controls and benchmark splicing modulators. Without these experiments it isn't clear how well the authors' system actually works, which will impede its uptake by other researchers. Therefore, this study requires significantly more development before publication.

We have performed RNA-seq analysis on CASFx and PUF-ESF (Fig 4c). Two potential off-target events were identified in samples transfected with CASFx-1 while seventy-nine off-targets were identified for PUF-ESF. Furthermore, out of the statistically significant off-target events called by MATS, CASFx affect potential off-target exons with up to 14.3% change in inclusion level $|\Delta\Psi|$ while off-targets by PUF-ESF are more severely affected, with up to >50% $|\Delta\Psi|$ inclusion changes.

Reviewer #2 (Remarks to the Author):

The authors of this manuscript invented CASFx, artificial splicing factors by composing domains of RESPR and RNA-binding proteins RBFOX1/RBM38. The authors guided the chimera splicing factor to ISSs located in the intron of SMN2 and E7, and examined the function of the artificial splicing factors by RT-PCR. They further developed this to inducible CASFx (iCASFx) which can be induced by Rapamycin. Although their idea seems to be attractive, the data are too primitive to convince me. More careful study would be required before the publication as pointed below.

1) In Fig 1, RBFOX1N of the chimera splicing factor may activate E7 recognition just by nonspecifically interfering the binding of hnRNPs to ISSs. The authors should confirm that the RBFOX1N chimera splicing factor activated E7 by binding to the ESS in E7 and that which part of RBFOX1N is required for the activation.

As shown in Fig 1b and Fig 2b, exon activation is not due to steric interference of hnRNP binding to ISSs since RBFOX1 and RBM38 fusions are required for such activity. We appreciate the suggestion about confirming the specific regulation of RBFOX1 effector as well as identifying the functional domains for exon activation. We have constructed and tested several truncation mutants of RBFOX1 domain. RBFOX1 N-terminal is dispensable but is less efficient than the RBFOX1N-dCasRx-C fusion. Further truncation or “division” of the C-terminal domain abolish activity (Supplementary Fig 1b). These experiments further support the requirement of splicing factor activity for E7 inclusion induced by CASFx.

2) In Fig2, dCasRx alone promotes E7 skipping suggesting that dCasRx-binding itself may interfere with U1 or U2 binding at either the donor site or acceptor site, because E7 is very short (only 44 nucleotides).

We agree with this explanation. The previous study of Silvana Konermann has demonstrated that dCasRx binding to would be sufficient for targeted exon splicing perturbation. Based on their result, dCasRx showed exon skipping activity when bound to either splice acceptor site or splice donor site.

3) In Fig3, it is not surprised that binding of dCasRx to the splicing acceptor site suppressed the CX exon inclusion, though the RT-PCR data of RG6 is not so clear in comparison with SMN2. The authors should show the data of dCasRx without RBFOX1N as control. In addition, gRNA of DR-SMN2-3-DR in panel B should be gRNA of DR-SMN2-2-DR.

We have repeated the experiment and gotten updated gel image for RG6 RT-PCR. We also added quantitative measurement of splicing pattern changing via real-time qPCR. In Fig 2 we already confirmed that both dCasRx and RBFOX1N-dCasRx-C were able to induce exon skipping when binding onto the targeted exon. Also, we corrected the mislabeling of gRNA in Fig 3A.

4) Though Rapamycin-dependent splicing regulation is an interesting idea, it is not sure that the activity of RBFOX1/RBM38 is really regulated by Rapamycin. They need to show the data of an FRB-conjugated protein as control.

We have repeated the experiments with more controls including an FRB-GFP construct as suggested by reviewers, and we proved that rapamycin, dCasRx target binding and RBFOX1/RBM38 effector domains are all required for the inducible splicing modulation (Fig 5b). We also have quantitative measurement via quantitative PCR to strengthen the conclusion.

5) The authors should show the sequence data of target of sgRNA in exon 7 and intron 7. I could not find them at the addgene site.

We have added sequence information in supplementary table.

6) All the data just come from the transient transfection to HEK293T and RT-PCR. The authors should confirm their data in other cell lines. In addition, they should test whether their chimera splicing factors can regulate splicing of endogenous pre-mRNAs .

To show that CASFx can work in different cell types, we have conducted splicing experiments on SMN2 and NUMA1 minigenes in two additional cell lines, HeLa, and U2OS (Supplementary Fig 2b, 2c). For endogenous genes, as mentioned above, we have additional data using CASFx to modulate splicing of endogenous SMN2 in SMA patient fibroblasts (Fig 6).

7) They should examine the function of their chimera splicing factors targeting splicing of RBFOX- and RBM38-dependent exons. By replacing the cis-element with sgRNA would be contribute to prove the function of their artificial splicing factors.

We have targeted NUMA1 cassette exon, which is known to be regulated by RBFOX splicing factors, in the context of HeLa and U2OS cells with endogenous RBFOX knocked down, demonstrating the function of chimeric splicing factors in regulating RBFOX-dependent exons (Supplementary Fig 3).

8) If the authors will show that their chimera splicing factors can regulate any alternative splicing using in vitro splicing system, they would convince us more about their chimera splicing factors.

Though the in-vitro application of CASFx would be interesting, the main purpose of our current study is to develop it as a toolkit for cellular study. The experiments of testing CASFx in different cell types (HEK293T, HeLa, U2OS, SMA patient fibroblast GM03813) further support the applicability of CASFx in different cellular contexts.

Reviewers' Comments:

Reviewer #1:

Remarks to the Author:

In this revised manuscript from Jillette and Cheng, the authors have now benchmarked their system against existing tools, tested it on endogenous genes, and looked for transcriptome-wide off targets. These experiments have provided a more solid basis for their claims and the work is improved because of this. Before this study is acceptable, however, further details and analysis of the RNA-Seq data need to be provided and an array of minor improvements still need to be made:

The RNA-Seq data.

The experimental details for this are very sparse, a detailed description needs to be provided. For example, it is essential that all RNA-Seq experiments are performed in triplicate – was this the case for these data? The SRA accession was not yet active to check this and it was not mentioned in the manuscript.

Which RNAs were affected in the analysis? Was the target, SMN2, detected in the analyses, only two significant changes are shown in Fig. 4ci and these are described as off-targets? The most (perhaps top 10) altered exons should be labelled, based on the gene and exon number they are produced from.

Further analysis of these data could provide interesting information. Were there sequence elements in common in the affected off-target exons?

CASFx versus ESF.

In Fig. 4b the comparison between a mix of the three ESFs and the all-in-one CASFx systems should be clarified in the results text. Since 6 ug of DNA is transfected in these experiments, the fact that the ESFs are transfected as three separate plasmids compared to a single all-in-one CASFx plasmid means that the actual molar amount of CASFx is higher than that of the ESFs. Hence the averaging effect seen for the ESF mix and the additive effect seen for all-in-one CASFx. This needs to be explained in the text so that readers can make an unbiased assessment of the comparison.

Scientific language.

The language throughout the manuscript needs to be tightened up. Random examples:

'However, Scalable tools for precisely and reversibly "writing" alternative splicing is lacking.' should be 'However, scalable tools for precisely and reversibly "writing" alternative splicing are lacking.' and all scientific units should be separated from the values by a space in the methods and the figures.

Reviewer #2:

Remarks to the Author:

This is a resubmitted manuscript of CRISPR artificial splicing factors (CASFx) composed by CRISPR and RNA-binding proteins RBFOX1/RBM38. They responded to most of my questions and requests and the manuscript was very much improved from the previous version. However, I further request the revision of following points.

In the previous comment 1), I suspected the function of RBFOX1N domain. As they used RRM of RBFOX1, I requested them to confirm whether the RNA-binding activity of RBFOX1N is required for the splicing regulatory function of CASFx. Although they added the data of deletion mutants of CASFx-1 in Sup fig 1 and showed the requirement of RBFOX1N, the role of this is still not clear. If RNA-binding activity of RBFOX1 is critical for the function of the CRISPR-based artificial splicing factors, construct of the chimeric splicing factors requires the design dependent on the target exons. This is the reason why I stick to this point.

Panel a of Fig 5 seems to be same as panel A of Fig 4 of the previous version, though the data in panel b were totally changed. Authors should renew the panel a, because data of RBM38-FRB or FRB-RBM38 are not shown in the panel b in the new version of the manuscript. In addition, they should mention about control and RG6-SA in the legend of Fig 5.

Response to reviewers' comments (NCOMMS-18-36967)

First of all, we appreciate the reviewers' time and feedback. We have added substantial data to support and improve the manuscript according to the valuable suggestions offered by the reviewers. Here we would like to explain changes to the manuscript point-by-point below:

Reviewer #1 (Remarks to the Author):

In this revised manuscript from Jillette and Cheng, the authors have now benchmarked their system against existing tools, tested it on endogenous genes, and looked for transcriptome-wide off targets. These experiments have provided a more solid basis for their claims and the work is improved because of this. Before this study is acceptable, however, further details and analysis of the RNA-Seq data need to be provided and an array of minor improvements still need to be made:

1. The RNA-Seq data.

The experimental details for this are very sparse, a detailed description needs to be provided. For example, it is essential that all RNA-Seq experiments are performed in triplicate – was this the case for these data? The SRA accession was not yet active to check this and it was not mentioned in the manuscript.

Which RNAs were affected in the analysis? Was the target, *SMN2*, detected in the analyses, only two significant changes are shown in Fig. 4ci and these are described as off-targets? The most (perhaps top 10) altered exons should be labelled, based on the gene and exon number they are produced from.

Further analysis of these data could provide interesting information. Were there sequence elements in common in the affected off-target exons?

Thank you for providing suggestions for off-target analysis. In the previous submission, the RNA-seq data for all-in-one CASFx-1 was performed with single sample. Due to the COVID-19 situation, we were not able to repeat experiments and sequencing for all-in-one CASFx1 in a timely manner. Therefore, we have analyzed off-target effects for CASFx-1, CASFx-3 and ESF mix using our available RNA-seq data performed in replicates (Fig 4c).

In the new off-target effect analysis, we added gene symbols to the off-target exons identified in CASFx-1 and CASFx-3 samples (Fig 4c i and ii). For the 59 identified off-target exons in ESF mix samples (Fig 4c iii), we listed the information of these exons in supplementary table 3. We have also revised method descriptions towards RNA-seq experiment and analysis.

We have added the mismatch calculation of *SMN2*-gRNAs and off-target exons. For each off-target exon found in CASFx-1 and CASFx-3, the sites with best match to gRNAs were shown in supplementary figures 2 and 3.

A Bioproject accession number has been added. The data is also now available for download.

2. CASFx versus ESF.

In Fig. 4b the comparison between a mix of the three ESFs and the all-in-one CASFx systems should be clarified in the results text. Since 6 ug of DNA is transfected in these experiments, the fact that the ESFs are transfected as three separate plasmids

compared to a single all-in-one CASFx plasmid means that the actual molar amount of CASFx is higher than that of the ESFs. Hence the averaging effect seen for the ESF mix and the additive effect seen for all-in-one CASFx. This needs to be explained in the text so that readers can make an unbiased assessment of the comparison.

The total amount of all the plasmids is 600 ng for each transfection sample. For ESF mix sample, equally split amount (194 ng) of all three ESF plasmids were co-delivered into cells. Therefore, total amount of effectors transfected into these two samples are the same. We have edited the method section to clarify.

3. Scientific language.

The language throughout the manuscript needs to be tightened up. Random examples: 'However, Scalable tools for precisely and reversibly "writing" alternative splicing is lacking.' should be 'However, scalable tools for precisely and reversibly "writing" alternative splicing are lacking.' and all scientific units should be separated from the values by a space in the methods and the figures.

We have made corrections to our scientific language and units.

Reviewer #2 (Remarks to the Author):

This is a resubmitted manuscript of CRISPR artificial splicing factors (CASFx) composed by CRISPR and RNA-binding proteins RBFOX1/RBM38. They responded to most of my questions and requests and the manuscript was very much improved from the previous version. However, I further request the revision of following points.

1. In the previous comment 1), I suspected the function of RBFOX1N domain. As they used RRM of RBFOX1, I requested them to confirm whether the RNA-binding activity of RBFOX1N is required for the splicing regulatory function of CASFx. Although they added the data of deletion mutants of CASFx-1 in Sup fig 1 and showed the requirement of RBFOX1N, the role of this is still not clear. If RNA-binding activity of RBFOX1 is critical for the function of the CRISPR-based artificial splicing factors, construct of the chimeric splicing factors requires the design dependent on the target exons. This is the reason why I stick to this point.

The previous study by Shuying Sun (PMID: 22184459) has demonstrated that flanking domains of RBFOX1 (N, C) lacking the entire RNA recognition motif (RRM) are sufficient for activating alternative exon inclusion when tethered to a target RNA. In CASFx-1, we also replaced the entire RRM (aa118-189) of RBFOX1 by dCasRx, so the intrinsic RNA-binding activity of RBFOX1 is abolished and replaced by dCasRx. Besides, the SMN2 minigene lacks RBFOX1 binding motifs. Therefore, RNA-binding activity of RBFOX1 is not involved in the function of CASFx-1.

Panel a of Fig 5 seems to be same as panel A of Fig 4 of the previous version, though the data in panel b were totally changed. Authors should renew the panel a, because data of RBM38-FRB or FRB-RBM38 are not shown in the panel b in the new version of

the manuscript. In addition, they should mention about control and RG6-SA in the legend of Fig 5.

We have corrected Fig 5a. We have repeated the experiment for Fig5b and replaced with the new qPCR plot as well as gel image. We added the discussion about RG6-SA and control gRNA in the result part of this figure.

REVIEWERS' COMMENTS:

Reviewer #1 (Remarks to the Author):

In this further revised version of their "CRISPR Artificial Splicing Factors" manuscript the authors have refined the work but have still not addressed my two main points in a satisfactory manner.

The RNA-Seq data.

The authors state that it is difficult to perform the RNA-Seq experiments in triplicate due to the COVID-19 situation. Although understandable, it is critical that the science produced during this period stands the test of time, as such I wonder whether it could be worth waiting for these data. My point regarding SMN2 is that, as the target gene, it should be significantly affected in the analyses shown in Fig. 4c. Is SMN2 not shown for some reason or is it not significantly affected? Either way it should be colored and labelled as a data point on these graphs.

CASFx versus ESF.

My point here also seems to have been misunderstood. Naturally the same mass of DNA was delivered in each case, however, I'm talking about the molar amounts of each gene delivered. If the proteins were tagged a comparison by western blotting of the protein levels might be possible but at this stage we can only infer from the amounts of DNA transfected. The ESFs are transfected as three separate plasmids compared to a single CASFx plasmid. Therefore, the actual molar amount of CASFx is different than that of the individual ESFs. As I previously noted, this might explain the averaging effect seen for the ESF mix and the additive effect seen for the all-in-one CASFx. This point should be made in the manuscript text so that readers are aware of this when they consider the comparison between the ESF mix and the all-in-one CASFx treatments.

We would like to thank the reviewers for their comments and provide response below:

Reviewer #1 (Remarks to the Author):

In this further revised version of their “CRISPR Artificial Splicing Factors” manuscript the authors have refined the work but have still not addressed my two main points in a satisfactory manner.

The RNA-Seq data.

The authors state that it is difficult to perform the RNA-Seq experiments in triplicate due to the COVID-19 situation. Although understandable, it is critical that the science produced during this period stands the test of time, as such I wonder whether it could be worth waiting for these data. My point regarding SMN2 is that, as the target gene, it should be significantly affected in the analyses shown in Fig. 4c. Is SMN2 not shown for some reason or is it not significantly affected? Either way it should be colored and labelled as a data point on these graphs.

We have colored and labelled the RNA-seq values of endogenous SMN2 in the plot. We have added the explanation why the endogenous SMN2 splicing cannot be accurately determined: No significant changes in endogenous SMN2 splicing were detected by RNA-seq (Fig 4c). It is because the high sequence homology between SMN1 and SMN2 and the constitutive inclusion of SMN1 exon 7 in HEK293T preclude accurate quantification of endogenous SMN2-E7 splicing level.

CASFx versus ESF.

My point here also seems to have been misunderstood. Naturally the same mass of DNA was delivered in each case, however, I’m talking about the molar amounts of each gene delivered. If the proteins were tagged a comparison by western blotting of the protein levels might be possible but at this stage we can only infer from the amounts of DNA transfected. The ESFs are transfected as three separate plasmids compared to a single CASFx plasmid. Therefore, the actual molar amount of CASFx is different than that of the individual ESFs. As I previously noted, this might explain the averaging effect seen for the ESF mix and the additive effect seen for the all-in-one CASFx. This point should be made in the manuscript text so that readers are aware of this when they consider the comparison between the ESF mix and the all-in-one CASFx treatments.

We have made the readers aware of the potential caveat of the difference in ESF mix and CASFx transfection as follows: One caveat for such interpretation, however, is that the higher off-target effects of PUF-ESF could also result from the relatively higher molar amount of the three PUF-ESFs transfected as separate plasmids.